# Prehistorical and historical declines in Caribbean coral reef accretion rates driven by loss of parrotfish

Katie L. Cramer[1,2], Aaron O'Dea[2], Tara R. Clark[3], Jian-xin Zhao[3] & Richard D. Norris[1]

Caribbean coral reefs have transformed into algal-dominated habitats over recent decades, but the mechanisms of change are unresolved due to a lack of quantitative ecological data before large-scale human impacts. To understand the role of reduced herbivory in recent coral declines, we produce a high-resolution 3,000 year record of reef accretion rate and herbivore (parrotfish and urchin) abundance from the analysis of sediments and fish, coral and urchin subfossils within cores from Caribbean Panama. At each site, declines in accretion rates and parrotfish abundance were initiated in the prehistorical or historical period. Statistical tests of direct cause and effect relationships using convergent cross mapping reveal that accretion rates are driven by parrotfish abundance (but not vice versa) but are not affected by total urchin abundance. These results confirm the critical role of parrotfish in maintaining coral-dominated reef habitat and the urgent need for restoration of parrotfish populations to enable reef persistence.

[1] Center for Marine Biodiversity and Conservation, Scripps Institution of Oceanography, UC San Diego, La Jolla, California 92093, USA. [2] Smithsonian Tropical Research Institute, Box 0843-03092 Balboa, Republic of Panama. [3] Radiogenic Isotope Facility, School of Earth Sciences, The University of Queensland, Brisbane, Queensland QLD 4072, Australia. Correspondence and requests for materials should be addressed to K.L.C. (email: katie.cramer@gmail.com).

Caribbean coral reefs are among the most degraded reefs on the planet[1,2]. Since systematic reef monitoring began in the 1970s, researchers have documented a dramatic 'phase shift' on Caribbean reefs whereby habitats previously dominated by reef-building corals (in many locations, primarily branching corals from the *Acropora* genus) are now dominated by macroalgae and low-relief corals tolerant of lower water quality (higher turbidity and nutrient) conditions[3–5]. This phase shift followed disease outbreaks that killed *en masse* the *Acropora* corals and the sea urchin *Diadema antillarum* in the early 1980s as well as coral bleaching outbreaks that became widespread in the late 1980s (refs 6–8). The appearance and intensification of coral disease and bleaching epidemics in the Caribbean and elsewhere have been linked to elevated sea surface temperatures from global climate change[9–12], algal overgrowth from overexploitation of herbivorous reef fishes[1,13,14] and increases in land-based runoff[15,16].

While dramatic changes have been observed over the most recent decades, historical, archaeological and paleontological data reveal the antiquity of human disturbances to Caribbean reefs. Fishing and land clearing activities appear to have been altering reef communities and environments for centuries to millennia: exploitation of Caribbean reef megafauna, fishes and invertebrates began centuries before the arrival of Columbus in 1492 (refs 2,17–22), and early intensive agricultural activities had degraded coral and mollusk communities on some reefs one to four centuries before disease and bleaching outbreaks[23–25]. However, the prehistorical and historical record of reef ecological change is primarily qualitative, poorly temporally constrained and rarely linked to contemporaneous human impacts to reef ecosystems, preventing a mechanistic explanation for recent and past declines that distinguishes between symptoms (bleaching and disease epidemics) and drivers (fishing, land-clearing, pollution and climate change).

Although there is scientific consensus that local and global anthropogenic stressors have negatively impacted Caribbean reefs, debate is ongoing about the initial timing and hence dominant drivers of recent change. While some studies document one or more centuries of decline in reef communities[2,18,20,21,24], other core-based studies of coral communities found little change in corals until the 1980s (refs 4,26). This debate has been confounded by the imprecision of radiocarbon dates for the historical time period[27]. The assessment of causes of change has also been hindered by the synergistic nature of anthropogenic stressors, which are currently impacting reefs simultaneously. For example, recent coral disease epidemics have been identified as the major cause of reef coral decline[8], but coral disease is exacerbated by elevated macroalgal abundance[28–30] which is a result of (1) longstanding local disturbances such as overfishing of keystone herbivores including parrotfish[1,5,11] and inputs of nutrients and pollutants onto reefs from agricultural and industrial activities[16,31–33] and (2) more recent acute and geographically extensive events such as the *Diadema* dieoff and coral bleaching[12,34]. The debate about the relative importance of historical and local versus recent and regional or global anthropogenic causes of reef declines (and the magnitude of their interactive effects—see ref. 35) has important management consequences, as the complexity of approaches is increased with the geopolitical scale of anthropogenic drivers[5,36,37].

To help resolve the role of herbivory loss in Caribbean reef ecosystem declines, we produce a continuous, high-resolution reconstruction of change in reef fish, coral, and urchin communities over the past three millennia by analysing the recent remains ('subfossils') of fish teeth, coral fragments and urchin spines preserved within reef sediment cores collected at three sites in Bocas del Toro, Caribbean Panama. Whereas previous studies of historical reef change in the Caribbean have had relatively poor temporal resolution due to the imprecision of radiocarbon dating for recent material[4,24–27], we utilize uranium-thorium (U–Th) dating to develop a high-resolution chronology of change in fish, coral and urchin composition and reef accretion rates. Coupled with a recently developed technique that assesses time-delayed causal relationships, convergent cross mapping (CCM)[38–40], our approach allows us to (1) produce the first historical reconstruction of reef fish communities from abundant fish teeth subfossils, (2) track changes in reef accretion rates from a continuous millennial-scale record of coral-dominated reef sediment accumulation and (3) quantify the causative relationship between reef accretion rate and the abundance of major reef herbivores—parrotfish and urchins.

At each site, declines in accretion rates and parrotfish abundance were initiated in the prehistorical or historical period. CCM analyses revealed that accretion rates were driven by parrotfish abundance (but not vice versa) but were not affected by total urchin abundance. These results confirm the critical role of parrotfish in maintaining coral-dominated reef habitat and the urgent need for management actions to maintain and restore parrotfish populations to enable reef persistence in the Caribbean.

## Results

**Collection of reef matrix cores.** Cores were collected from three sites within the semi-enclosed lagoon Almirante Bay that span a gradient of influence from land-based runoff[41]. Two sites, Airport Point and Cayo Adriana, are fringing reefs located along the southern coast of Colon Island ∼10 and 18 km from the mainland coast of Almirante Bay, respectively, while Punta Donato is a patch reef located ∼2 km from the western mainland coast, an area of industrial-scale banana agriculture since the late 19th century[33] (Fig. 1). A previous analysis of change in coral and molluscan subfossil assemblages spanning from approximately 1900 AD to present from pits excavated at Punta Donato revealed that this site first experienced significant declines in reef water quality at least a century ago[24–25].

**Reef matrix composition and U–Th chronology.** Reef cores are primarily composed of a dense matrix of coral skeleton and mollusk shell fragments (>2 mm) within a sandy-muddy matrix of carbonate grains. The 27 [230]Th ages obtained from all six cores reveal that these cores span the period 1239 ± 15 BC to 1984 ± 3 AD. Corrected [230]Th age errors (2σ) range from ± 3 to 15 years (see Supplementary Table 1). Nearly all [230]Th ages are in chronological order, but one age reversal occurs in the top meter section of each of the well-dated cores from Airport Point and Cayo Adriana, indicating a notable slowdown of reef accretion rates approximately 1,000 years ago at these sites (Fig. 1). Reef accretion rates at Punta Donato peak from ∼1000–1500 AD and slow from ∼1500 AD to the tops of the cores. The bottom sections of both Punta Donato cores are composed of quartz sand that was initially colonized by bivalves before corals were established on the shell layer; these cores capture the full lifespan of this reef, from initial development (characterized by slow initial reef accretion) to cessation. Because we cored dead rubble zones adjacent to living reef, [230]Th ages from the top of three cores are not modern. While cores from the patch reef at Punta Donato extend into the 1950s and 1980s, those from the fringing reefs at Airport Point and Cayo Adriana terminate much earlier: both Airport Point cores terminate near 1200 AD, while the Cayo Adriana cores terminate at ∼1000 AD and ∼1925 AD

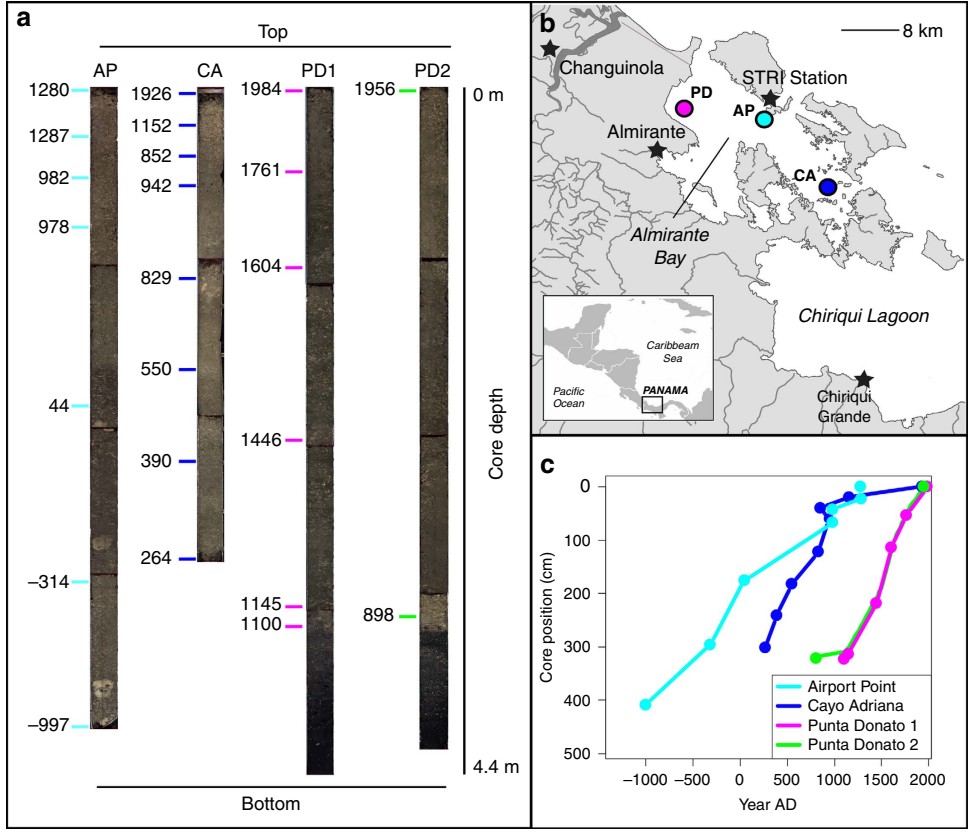

**Figure 1 | Reef accretion rates over last 3,000 years in Bocas del Toro, Panama.** (**a**) Reef matrix cores (n=4) analysed for subfossil and sediment composition and U–Th dates (n = 23) obtained (in year AD) along length of cores. AP = Airport Point, CA = Cayo Adriana, PD1 and PD2 = replicate cores from Punta Donato. (**b**) Map of coring locations. Stars indicate population centers, including Smithsonian Tropical Research Institute (STRI)'s research station. Turquoise = AP, blue = CA, pink and green = PD1 and PD2, respectively. (**c**) Age-depth plot showing reef accretion trends Age reversals excluded from linear interpolations of age estimates; rates for PD2 were assumed to be equivalent to those for same core position in PD1.

(Supplementary Table 1). A linear fit through all the dates in the well-dated Cayo Adriana core suggest that accretion slowed dramatically around 1300–1400 AD (Fig. 1).

The cessation of reef growth at the two fringing reefs, despite areas of active coral growth directly downslope of our coring locations and within 0.5 m of the water surface, suggests that the zones in which we cored may have transitioned from actively accreting to non-accreting environments due to changes in local hydrography related to changing sea level[42]. We therefore analysed the subfossil and sediment composition from the most recent core from each of the fringing reefs and both cores from the Punta Donato patch reef to provide a high-resolution record of change over the past ~3,000 years, with Airport Point representing the prehistorical period (997 BC–1280 AD), Cayo Adriana representing the prehistorical-early industrial period (264 AD–1926 AD), and Punta Donato representing the prehistorical-modern (post-industrial) period (1100 AD–1984 AD for core 1 and 898 AD–1956 AD for core 2; Fig. 1). Linking accretion rates between the well-dated and replicate core from Punta Donato (PD1 and PD2, respectively) shows a ~200 year mismatch between estimated and observed bottom values, indicating a lower reef accretion rate at PD2 (Supplementary Table 1).

**Long-term trends in herbivore abundance and reef accretion.** The subfossil fish tooth assemblage is composed of caniniform, incisiform, molariform and hybrid morphotypes and fragments, with individual core samples containing 74 teeth on average

(range = 2–232). The coral assemblage is composed primarily of branching *Porites* spp. (mainly *Porites furcata*), *Agaricia* spp., *Madracis mirabilis* and *Acropora cervicornis*, and the urchin assemblage is is composed primarily of *Echinometra* spp., *Lytcehinus/Tripneustes* spp. and *Diadema antillarum* (Fig. 2). There is an overall decline in the absolute abundance of fish teeth and corals and an overall increase in urchins across the full time series (Fig. 3). Within individual cores, peaks in absolute abundance of fish teeth generally coincide with periods of high reef accretion rates, despite the shorter period of time for subfossils to accumulate in these samples. Similarly, minima in fish tooth abundance occur during periods of low reef accretion rates despite the longer period of time represented in these samples that would allow for a greater accumulation of subfossils.

Comparison of subfossil tooth morphotypes to our modern tooth reference collection reveals that 46% of all teeth belong to parrotfish (family Labridae), the most important herbivores on post-*Diadema* Caribbean reefs[43] and whose depletion on most reefs has been linked to the replacement of corals with macroalgae following the *Diadema* die-off[1,5]. The relative abundance of parrotfish decreases across the time series and region as a whole, with notable declines at individual sites beginning at ~100 AD at Airport Point, ~1000 AD at Cayo Adriana and ~1600 AD at Punta Donato, coinciding with declines in accretion rates and total tooth abundance for the latter two sites (Fig. 3). Relative abundance of parrotfish teeth is a reliable proxy of absolute parrotfish tooth abundance, as these measures are closely positively correlated (Supplementary Fig. 1).

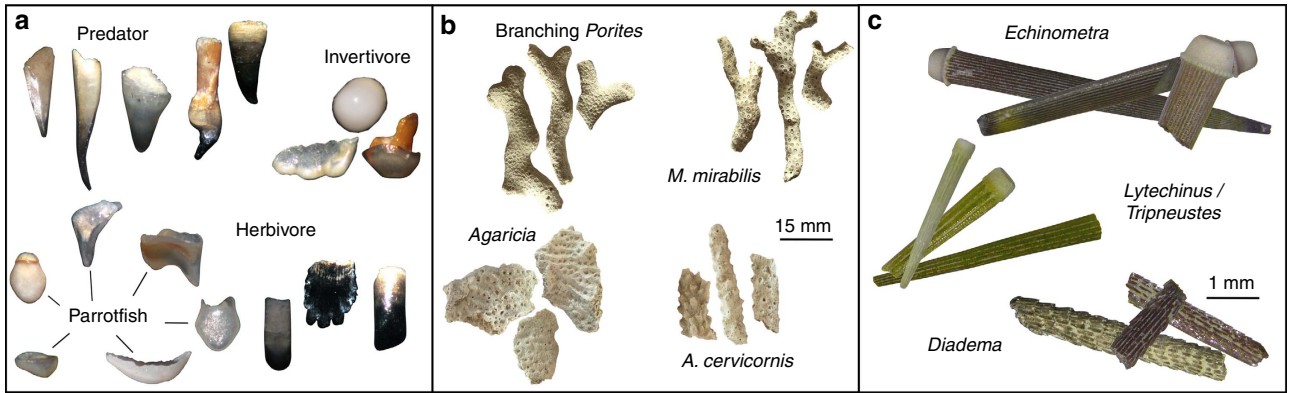

**Figure 2 | Major subfossil groups preserved in reef matrix cores.** (**a**) Fish teeth functional groups and variety of parrotfish tooth morphotypes. Teeth range from 500–63 μm in size but majority are 250–104 μm; images not to scale. (**b**) Common coral taxa identified from skeletal fragments. (**c**) Common urchin taxa identified from spines.

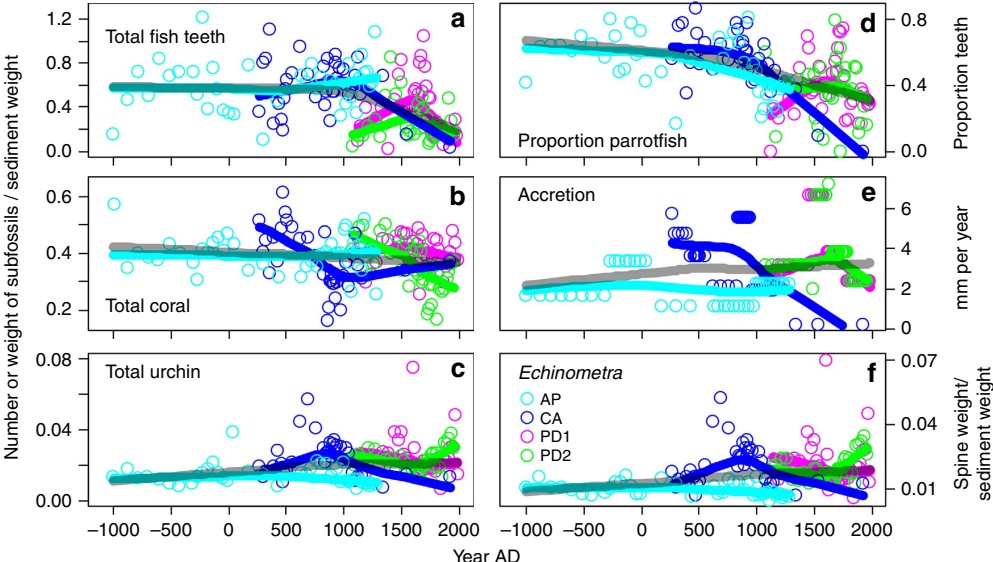

**Figure 3 | Millennial-scale trends in abundance of major reef subfossil groups and accretion rates.** (**a**) Total fish tooth abundance measured as number of individual teeth divided by dry weight of all sediment size fractions combined. (**b**) Total coral abundance measured as weight of coral fragments from > 2 mm sediment fraction divided by dry weight of > 2 mm fraction. (**c**) Total urchin abundance measured as weight of urchin spines divided by dry weight of 0.5–2 mm sediment fraction. (**d**) Proportion of all teeth belonging to parrotfish. (**e**) Reef accretion rate represented as mm of reef sediment accumulated per year. (**f**) Abundance of *Echinometra* urchins measured as total spine weight of genus divided by dry weight of 0.5–2 mm sediment fraction. Number samples = 154. Turquoise = Airport Point, blue = Cayo Adriana, pink and green = Punta Donato 1 and 2, respectively. Coloured lines are loess smoothed estimated trends for individual cores and semitransparent black lines are loess smoothed estimates for all cores combined; smoothing parameter = 0.9.

Branching *Porites* consistently dominates the coral assemblage within most cores and time periods, except for periods of dominance of staghorn coral *A. cervicornis* from 1000–1920 AD at Cayo Adriana, branching *M. mirabilis* from approximately 1500–1800 AD in both Punta Donato cores, and temporary and periodic dominance of lettuce coral *Agaricia* at Airport Point and Cayo Adriana (Supplementary Fig. 2). The urchin assemblage is consistently dominated by *Echinometra* (85% total spine weight), whose abundance closely tracks that of overall urchin abundance (Fig. 3). Typically seagrass- and reef-associated *Lytechinus/Tripneustes* urchins comprise 12% of overall spine weight, while the keystone herbivore *D. antillarum* is consistently rare across cores, comprising 3% of overall spine weight.

**Causal relationships between herbivory and reef accretion.** There is a significant causal relationship between reef accretion

and parrotfish abundance. However, this relationship is unidirectional: while parrotfish abundance (measured in relative or absolute terms) positively affects reef accretion rate, reef accretion rate has no causal effect on parrotfish abundance (Fig. 4). In contrast, we detect no causal relationship between total urchin abundance (measured as the proportion of sediment weight from the 0.5–2 mm size fraction comprised of urchin spines) and reef accretion (Fig. 4). When urchin taxa are considered separately, *Lytechinus/Tripneustes* has a significant positive causal effect on accretion rate, but neither *Echinometra* nor *Diadema* are causally related to accretion (Supplementary Fig. 3). Parrotfish abundance has a positive causal effect on the abundance of the dominant urchin taxon *Echinometra* but not *Diadema* or *Lytechinus/Tripneustes* (Supplementary Fig. 3). There is no causal relationship between the relative abundance of branching *Porites*, the only consistently common coral species across the cores during the last three millennia, and reef accretion

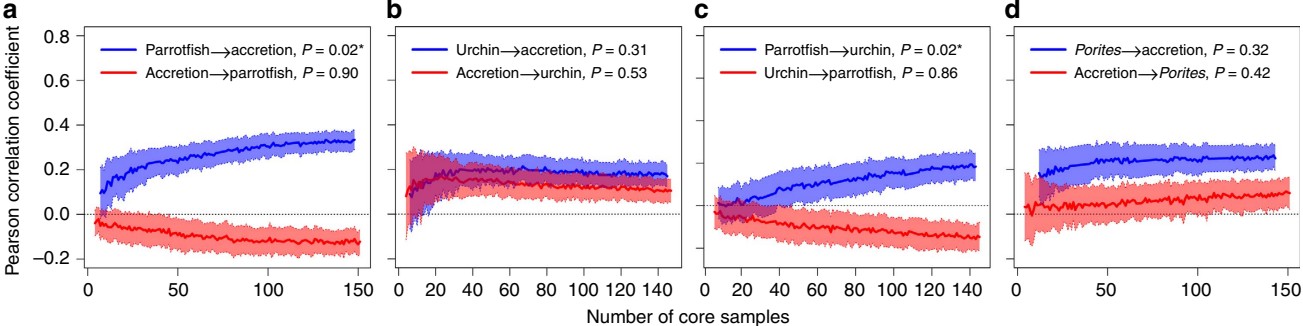

**Figure 4 | Analysis of causality between reef accretion and abundance of herbivores or dominant coral.** (**a**) Parrotfish relative abundance and reef accretion rate. (**b**) Urchin abundance and reef accretion rate. (**c**) Parrotfish relative abundance and urchin abundance. (**d**) Branching *Porites* coral relative abundance and reef accretion rate. Lines and shaded regions show mean ± s.d. from 100 bootstrapped iterations. Significant causal forcing (*) determined from bootstrap test with 100 iterations, and indicated when the Pearson correlation coefficient is significantly greater than zero for larger sample sizes (number of core samples, including all spatial replicates in the composite time series) and when correlation coefficient increases significantly with increasing number of core samples.

rates (Fig. 4). Diagnostic plots show assumptions of nonlinearity and nonrandomness were met for each variable pairing (Supplementary Figs 4 and 5).

## Discussion

The subfossils and sediments analysed from our reef matrix cores provide a 3,000-year continuous record of reef growth and herbivore abundance in Bocas del Toro that spans the period of intensifying human impacts to coastal marine ecosystems[18,19,44]. At the coring locations of each of our three sites, reefs show declines in accretion rates and fish abundance initiated in the prehistorical or historical period as they transition from actively accreting to non-accreting habitats. Reefs shift from systems with high overall fish and coral abundance with greater relative abundance of parrotfish during faster accretion periods to systems with reduced fish and coral abundance typified by fewer parrotfish during slower accretion periods.

Our analyses of the causal relationship between reef herbivore abundance and accretion rates demonstrate the essential role of herbivory in coral growth and abundance. During the prehistorical and historical periods, parrotfish abundance was a positive driver of reef accretion, indicating that their positive effects on coral growth via removal of benthic macroalgae outweigh their negative effects via corallivory and bioerosion[45,46] (although parrotfish may facilitate coral disease on eutrophied reefs—ref. 35). However, the lack of a causal effect of accretion on parrotfish abundance suggests that other factors, including fishing, are more important determinants of parrotfish population dynamics[47]. Our fish tooth record may indicate long-term declines in reef herbivory from prehistorical and historical artisanal fishing, as the highest relative abundances of parrotfish are found in the prehistoric records from Airport Point and Cayo Adriana (Fig. 3).

We did not detect a causal relationship between overall urchin abundance (consistently dominated by *Echinometra* throughout the past three millennia) and reef accretion, likely signifying that *Echinometra*'s positive effect on reef accretion via herbivory is of similar magnitude to their negative effect via intense coral boring activity[48,49]. In contrast, reef accretion is positively driven by the abundance of the subdominant *Lytechinus/Tripneustes* group, likely indicating their positive effects on accretion via herbivory are greater than any negative effects from bioerosion[50,51]. Surprisingly, the keystone herbivore and coral bioeroder *Diadema* was not found to be causally related to reef

accretion, likely a result of the consistent rarity of this urchin on the forereef slope zone of our coring sites. The multispatial CCM analyses reveal a positive causal effect of parrotfish abundance on total urchin abundance in general and *Echinometra* abundance in particular (Fig. 4, Supplementary Fig. 3). This effect indicates a mutualistic rather than competitive relationship between parrotfish and *Echinometra* that may be caused by parrotfish-mediated facilitation of (1) coral-dominated habitat and/or (2) palatable benthic algal communities preferred by this urchin. Reef accretion rates are not causally affected by the relative abundance of the most common coral species, branching *Porites*. While accretion rates would undoubtedly be affected by changes in the relative abundance of exceptionally fast growing species such as *Acropora cervicornis*[52], this species was only temporarily dominant at a single coring location, preventing a rigorous assessment of this relationship (Supplementary Fig. 2).

Although reef accretion and parrotfish abundance decline during the most recent ∼500–1,000 years within each core (Fig. 3), the timing of these transitions suggests differing causes of change. At Airport Point, accretion and parrotfish declines occur as early as two millennia ago and reef accretion stopped by ∼1280 AD. Fish declines are also evident ∼1000 AD at the Cayo Adriana reef but accretion also largely stops by ∼1300–1400 AD—broadly in the same period as the Airport Point reef. However, unlike Airport Point, the reef at Cayo Adriana continues very slow growth into the historic period. During the period between 0 AD and 1400 AD, low human population levels in Bocas del Toro presumably had relatively little environmental impact on reefs[33,53,54]. This timing and the persistence of living coral communities within just ∼5–10 meters of the coring locations at Airport Point and Cayo Adriana implicate non-human drivers of change, possibly related to alterations in reef hydrography resulting from local sea level fluctuation[42]. Notably, the reef at Punta Donato was becoming established just before the slowdown in accretion at the fringing reef sites, reinforcing the conclusion that the declines in accretion at the fringing reefs were not primarily due to human impacts but likely reflect the evolution of the hydrography of Almirante Bay.

In contrast, the more recent transition of the reef community at Punta Donato implicates human activities. Declines in fish and coral abundance and relative abundance of parrotfish become detectable between the mid-18th to mid-19th century and continue to the present, with very little living coral persisting at

this site today[24]. This period is one of relatively high human population density near Almirante Bay and increasing exploitation of coastal marine resources from indigenous inhabitants, non-local indigenous and European traders, pirates engaging in intensive harvesting of mega-herbivores including green sea turtles and manatees, and land clearing for industrial-scale banana agriculture[33,54,55]. The similar observed ecological trajectories of change across both historic and prehistoric reef sites suggest that reef ecosystem deterioration follows the same pattern whether caused by anthropogenic activities or natural events—historical local human activities appear to have unraveled reefs at a scale similar to past large-scale hydrological changes.

Factors associated with the preservation and accumulation of reef sediments and subfossil assemblages could have affected our results. Ages and accretion rates could be affected by uncertainties in age model estimation, likely primarily caused by linear interpolations between samples with dated coral pieces and/or uncertainties in precise core depth of individual samples from sediment compaction during or after the coring process (on average, post-coring settlement reduced core lengths by 0.8 m, or by 18% of their original length). Reef accretion rates declined dramatically in the top 1 m of the Airport Point and Cayo Adriana cores and each included one age reversal (which we excluded from the age models). Therefore, age estimations near these reversals are less certain than those in other samples. Although reef accretion rates were used as a proxy of coral growth, non-coral carbonate producing organisms also contributed to sediment production. However, 83% of the weight of the undigested sediments $> 500 \mu m$ (which account for $\sim 80\%$ of sample volume and 50% of total sediment weight on average), is made up of coral. Accretion rates could also have been affected by the degree of bioerosion or physical erosion which could create finer sediments that might be more easily removed from the reef matrix by current or wave action. However, because accretion rates are not related to sediment grain size (Supplementary Fig. 6), accretion estimates appear to be independent of bioerosion effects. Similarly, changes in reef accretion could result in less baffling of sediments between the coral fragments comprising the reef framework, possibly affecting the total abundance of smaller subfossils such as fish teeth. However, these changes would not be expected to significantly affect the taxonomic composition of fish teeth or urchin spines, as taxa within each of these groups contain subfossils of similar size. Smaller grains $< 2 mm$ (including fish teeth and urchin spines) could have been winnowed through cores via bioturbation or the coring process, moving them deeper along the core length than larger coeval particles (corals), leading to mismatch in ages between corals and smaller fossils. However, the tight temporal coupling between reef accretion rates (based off ages of coral fragments) and accumulation rates and community composition of fish teeth and urchin spines indicate minimal vertical reworking or winnowing occurred. Finally, the subfossil tooth record likely overemphasizes the contribution of parrotfish to the total living reef fish community, as the dentition of many parrotfish species includes a multi-element tooth battery, scraping and excavating feeding behaviours cause high tooth turnover rates and parrotfish teeth are more likely to be deposited in reef sediments compared with teeth from higher trophic level fishes that frequently migrate to non-reef habitats. However, as these factors would not be expected to vary over time, our tooth record is appropriate for tracking temporal change in parrotfish abundance.

Our analysis of millennial-scale change in reef communities in Bocas del Toro suggests that historical fishing may have been significantly affecting Caribbean reefs for over two centuries,

initiating ecosystem declines from which they have not recovered. The clear positive causal effect of parrotfish abundance on reef accretion rate indicates that modern Caribbean reefs that have largely been depleted of these herbivores may be locked into an alternative stable state of macroalgal dominance[56], and that positive accretion may cease on many reefs if parrotfish abundance remains low. Our subfossil data conclusively demonstrate that a significant and immediate reduction of fishing on parrotfish is necessary to enable coral recovery and persistence in the Caribbean.

## Methods

**Collection and processing of reef matrix cores.** To prevent damaging living corals, we collected cores from rubble zones on the reef slope adjacent to living coral colonies. At each site, two replicate cores 10.1 cm in diameter and ranging from 3–5.5 m in length were extracted $\sim 10$–30 m apart at 6–7 m water depth using using a self-contained underwater breathing apparatus (SCUBA) and a combination of push-coring and vibra-coring techniques. Cores were split lengthwise, and the working half was sliced into 5 cm increments (referred to as samples throughout the paper). Subfossils were analysed from every sample from the top 1 m of each core to produce a high-resolution record of change from the most recent period, while every third sample was analysed from deeper parts of the core to produce a moderately high-resolution record from the pre-historical period. This scheme resulted in the analysis of 36–41 samples from each core. Samples were dried and weighed, then wet sieved at 2 mm, 500, 250, 104 and 63 μm fractions to facilitate sorting and counting different taxonomic groups.

**Reconstructing reef ecosystem state.** We tracked changes in the abundance and taxonomic or functional composition of reef fish, corals and urchins through time. Reef accretion rate, a measure of carbonate production by hermatypic corals as well as other carbonate-producing organisms, was computed by determining sedimentation rates based on our age models (time/5 cm increment). Fish teeth were analysed from all five sediment size fractions. To remove small carbonate grains from acid-insoluble tooth subfossils, we dissolved the three sediment size fractions $< 500 \mu m$ in 10% acetic acid. Parrotfish teeth (family Labridae) were identified from gross tooth morphology and with the assistance of a tooth and jaw reference collection that we developed from dissection and photography of oral and pharyngeal jaws of 287 positively identified Caribbean reef fish species that represent all Caribbean reef families and most genera. Modern fish specimens were obtained from the Fish Collection of the Smithsonian Institution's National Museum of Natural History and fish markets in Bocas del Toro and Colón, Panama. Our photographic reference collection is at http://ichthyolith.ucsd.edu/.

Coral and urchin subfossils were assigned to species or generic level. Coral fragments from the $> 2 mm$ sediment fraction were weighed and identified to common taxa (*Acropora cervicornis*, *Agaricia* spp., branching *Porites* spp., *Madracis mirabilis*). Urchin spines were picked from the 0.5–2 mm fraction and separated into two typically reef-associated taxa (*Diadema antillarum*, *Echinometra* spp.) and one typically seagrass- or reef-associated group (*Lytechinus* and *Tripneustes* spp.) and counted and weighed. Total urchin abundance was also computed to track effects of overall urchin herbivory on reef accretion rates.

**Assessing timing of ecological change.** A high-resolution chronology of reef ecosystem change was obtained by U–Th dating coral fragments within the cores using a Nu Plasma multi-collector inductively-coupled plasma mass spectrometer (MC-ICP-MS) in the Radiogenic Isotope Facility at University of Queensland, following chemical treatment procedures and MC-ICP-MS analytical protocols described in ref. 57. For U–Th dating, each coral sample consisting of $\sim 150 mg$ fine sand-size chips that were carefully fragmented, $H_2O_2$-treated and then hand-picked under a binocular microscope to remove any trace detritus or grains with discolouration was spiked with a mixed $^{229}Th$–$^{233}U$ tracer and then completely dissolved in double-distilled concentrated $HNO_3$. After digestion, each sample was further treated with $H_2O_2$ to decompose trace amounts of organic matter (if any) and to facilitate complete sample-tracer homogenization. U and Th were separated using conventional anion-exchange column chemistry using Bio-Rad AG 1-X8 resin. After stripping off the matrix from the column using double-distilled 7N $HNO_3$ as eluent, 4N $HNO_3$ and 2%$HNO_3$ + 0.03% HF mixture was used to elute U and Th into 3.5 ml pre-cleaned test tubes, respectively. After screening U and Th concentrations in the U and Th fractions using their 1:100 dilute solutions on a quadrupole ICP-MS, the U and Th separates for the samples were then re-mixed in 2% $HNO_3$ to make $\sim 3 ml$ solution in a pre-cleaned 3.5 ml test tube. The mixed solution for each sample contained the entire Th fraction and a small percentage of the U fraction. The amount of U fraction to be added to the mixed solution was calculated based on the screening results and the MC-ICP-MS working sensitivity, aiming to achieve $\sim 5$ volts of $^{238}U$ signal. The remixed U–Th solution was injected into the MC-ICP-MS

through a DSN-100 desolvation nebulizer system with an uptake rate of $\sim 0.1\,\mathrm{ml\,min^{-1}}$. U–Th isotopic measurements were performed on the MC-ICP-MS using a detector configuration to allow simultaneous measurements of both U and Th isotopes. The $^{230}Th/^{238}U$ and $^{234}U/^{238}U$ activity ratios of the samples were calculated using the decay constants given in ref. 58. $^{230}Th$ ages were calculated using the Isoplot 3.75 Program[59]. $^{230}Th$ ages were corrected for non-radiogenic $^{230}Th$ contributions using a modelled two-component-mixing non-radiogenic $^{230}Th/^{232}Th$ value based on the equation in ref. 57 (also see Supplementary Table 1).

At each site, a highly constrained chronology was produced for one core where four $^{230}Th$ ages were obtained from the top 1 m of the core and one date from approximately every 0.5 m interval below that, yielding 6–8 ages per core (Fig. 1). From these cores, accretion rates were estimated using linear interpolation between each pair of ages. Age reversals were removed before interpolations. Only two $^{230}Th$ ages were obtained from the top and bottom of the replicate cores at each site, with the top age used to constrain the sedimentation rate to that of contemporaneous periods of the adjacent well-dated core.

**Data synthesis and interpretation.** To account for varying sediment quantities among individual samples, overall fish, coral and urchin abundances were determined from total number of teeth or total weight of coral fragments and urchin spines divided by the dry weight of the sediment size fraction(s) from which they were picked. Because proportional abundances of living communities are generally faithfully represented in subfossil assemblages[60], we computed relative abundances of parrotfish and coral species (relative to total tooth abundance and total coral weight, respectively). Urchin species abundance was calculated from the spine weight divided by the dry weight of the 0.5–2 mm sediment fraction from which they were picked.

Temporal trends in individual community components were assessed by plotting abundance values of individual samples, with trends assessed within each core and across all cores combined using non-parametric locally weighted regression ('loess')-smoothed trendlines[61]. The loess smoothing parameter was set to 0.9 for assessment of general trends across the full time series.

**Causality analyses.** To assess the presence and direction of causal relationships between herbivore abundance and reef accretion rates and between herbivore groups, we used CCM, a technique that compares the ability of time-lagged components of one process to estimate the dynamics of another. CCM tests for significant causal relationships by recognizing that the observed values of a forcing process should be significantly better explained by observed values of a response process than expected by chance, and the accuracy of that explanation should improve with increasing time series length[35–37]. We used a variant of CCM, multispatial CCM, which is appropriate for spatially replicated time series that individually contain short series ($\sim \leq 30$) of sequential ecological observations of systems that share similar dynamics[36], such as samples from our four individual cores.

Briefly, the algorithm proceeded in five steps: (1) using cross-validation, determine the optimal embedding dimension (E) that describes the number of time steps that best predicts the dynamics of each explored process (that is, parrotfish abundance, urchin abundance, accretion rate), (2) test for nonlinearity and stochastic noise to ensure that none of the processes were purely random and that stochastic noise was not so large that causal links could not be recovered. Stochasticity and nonlinearity were tested by visually assessing whether predictive power was reasonably high for short time steps and decreased with increasing prediction time, respectively[38,39], (3) calculate the ability of two processes to describe each other's dynamics using CCM by confirming that predictive skill (Pearson correlation coefficient) increased with greater number of historical observations (here, number of samples from a single core), (4) use bootstrapping with replacement to leverage spatial information to reshuffle the order of spatial replicates and calculate the correlation coefficient and (5) use nonparametric bootstrapping to test whether predictions indicated a significant causal relationship by determining whether calculated correlation coefficient was significantly greater than zero and whether it increased significantly with sample size. One hundred bootstrap iterations were performed for steps 4 and 5. All statistical analyses were conducted using the R software package[62]; causality analyses were conducted using the 'Multispatial CCM' package in R[39].

**Data availability.** The data that support the findings of this study are available from the corresponding author upon reasonable request.

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

## Acknowledgements

We thank F. Rodríguez, C. Angioletti, B. Degracia, M. Álvarez, E. Ochoa and T. Norris for help collecting sediment cores, A. Hangsterfer for help with transport and storage of core material, E. Sibert for help developing fish tooth isolation and sorting techniques, B. Oller, C. Carpenter, S. Buckley and M. Siltanen for processing reef sediments, A. Sanderson and D. Chen for help isolating and identifying fish teeth, C. Carpenter and L. Paulukonis for help isolating and identifying urchin spines, K. McComas, J. Williams, P. Hastings, D. Pitassy, H.J. Walker, M. Álvarez, F. Rodríguez, M. Pinzon Concepcion, A. Castillo for help developing the modern Caribbean reef fish tooth reference collection, H. Ye for guidance with convergent cross mapping analyses, D. Bellwood and S. Brandl for assistance identifying parrotfish teeth, R. Collin, G. Jácome and P. Góndola for help with field logistics, and J. Mate and Authority of Aquatic Resources of Panama for facilitating and providing collection permits. K.L.C. was supported by Smithsonian Institution MarineGEO and UC San Diego Frontiers of Innovation Scholars Postdoctoral Fellowships. The National System of Investigators of the National Secretariat for Science, Technology and Innovation of Panama supported A.O. Valerie and Bill Anders kindly helped to support fieldwork. This is contribution 12 from the Smithsonian's MarineGEO Network.

## Author contributions

K.L.C., R.D.N., and A.O'D. designed the study and collected data in the field, K.L.C., and R.D.N. processed core sediments and/or isolated and sorted subfossils, K.L.C. conducted statistical analyses and summaries of data trends, R.D.N. designed the coring system, T.R.C. and J.Z. created U-series chronologies for each core, K.L.C. wrote first draft of manuscript, and all authors contributed to revisions.

## Additional information

**Competing financial interests:** The authors declare no competing financial interests.

**Publisher's note**: 

