## [Peer Review File · Nature Communications]

Reviewers' comments:

Reviewer #1 (Remarks to the Author):

I read with attention the manuscript entitled "Prehistorical and historical decline of Caribbean reef ecosystems linked to loss of herbivores". This title is very appealing since it suggests a kind of causality. The study is also very timely since the predominant role of herbivores on the maintenance of healthy and productive coral reefs is still very controversial. The main paradigm is certainly the top-down control of the benthos by herbivores and particularly parrotfishes. Yet the other way around is also discussed, and even demonstrated in some cases, with a bottom-up control of herbivore abundance by the benthos (Russ et al. 2015 Mar Biol).

So I was quite excited to see how the authors used historical and even prehistorical records to provide a clear answer to this debate. I must say that I have no personal interest in this debate, I'm just seeking the truth through evidences.

I was very impressed by the originality, quantity and quality of the data. This sampling effort through time and across several groups already makes the paper a strong contribution. However I believe (i) that the manuscript could be simplified and be more straightforward, and (ii) that the conclusions are based on analyses that cannot correctly test the hypothesis and support a causal relationship.

My main criticism is that the authors did not use the appropriate framework and tools to seek causal relationships in time series. The so called "mirage" correlations are very common and I must say the simple plots or GLMs are not convincing. To overcome the pitfalls in the analysis of such trends I suggest to adopt the Granger causality approach (Sugihara et al. 2012 Science). For instance there is a way to test bidirectional vs. unidirectional coupling between variables which would be particularly relevant in the case of herbivore-coral relationships. The authors can also take advantage of their spatial replicates to really try to detect causal relationships (Clarke et al. 2015 Ecology).

The authors should also include more variables in the model like sea surface temperature, which may act on accretion, human density in the vicinity of the reefs, or predator abundance.

The manuscript is also too long in my opinion. The last part of the results, based on ordinations, seems useless.

In sum I believe that this study has the potential to become a key contribution in the field but a more solid analysis of the data is necessary to conclude any causal link and to convince the reader that the loss of herbivores is the cause and not the consequence of the degradation of coral reefs.

Reviewer #2 (Remarks to the Author):

This is an interesting paper that builds on previous work by the author on molluscan assemblage changes over time in the same region of the Caribbean. The main take home message about past (historical) declines in reef accretion rates is an important one, although as you will see below I have some major concerns about the interpretations being

made and the validity of the datasets given that they derive from individual core records - one from each of 4 geographically isolated reefs.

My main concerns are listed below.

1. The general headline statement about modern reefs and stable alternate states (last line of abstract and discussion). I am not sure this statement about alternative stable states is really accurate (or at least is one that can be supported with the data presented in THIS paper). Indeed, the findings of the paper, if correct, suggest that accretion rates and parrotfish and urchin abundance are correlated. It does not confirm that modern reefs are locked into an alternate stable state. This needs changing.

2. Introduction - Para 1, lines 4-5. These low relief corals are not only tolerant of higher turbidity/nutrients - indeed they occur on many disturbed reefs with little or no turbidity/elevated nutrient influence. I think it is also the case that they were quite common below/beneath branching coral stands at sites monitored prior to the Acropora collapse - it would be worth checking some of the 1970's/early 1980's data from places like Discovery Bay, and have been identified commonly on natural disturbance hiatus surfaces (see Scoffin's classic paper on hurricane deposited sequences).

3 Results 1 - I am not entirely convinced about your interpretation of accretion rate trends. Firstly, it is very concerning that accretion rates have been generated based on single cores - especially given that the central thrust of the story being presented hinges entirely on these rates. Accretion rates can commonly vary significantly within and across individual reefs and this raises some major questions to me about the dataset being presented. I note that only one site, Cayo Adriana, actually shows any really convincing reduction in accretion rates in the upper part of the cores (see figs 1 and 2). The other three sites look to have more steady state or even increasing rates over time? I note that the authors go on to suggest in the next paragraph that the old age of dates at the tops of some cores is consistent with a natural shut down of reef accretion potential, presumably as accommodation space declined. This would seem to me to be a highly likely interpretation. However, what this also means is it is very hard to assume that the ecological changes you infer are caused by recent environmental factors. It may well be (and I am not sure how you can rule this out - and it rather drives a wedge through the main arguments of the paper) that the up-core changes you infer in all of the cores where actually caused by a natural shift in reef ecology as the reef shallowed i.e., one would expect the types of corals and benthic taxa (especially foraminifera), to change as a reef grows vertically because the habitats themselves will also be changing. I am not sure why this transition under natural shallowing conditions has not been considered.

Results 2 - In terms of the measured abundance of fossils in the cores this obviously represents a significant amount of research time and there are some nice plots showing abundance versus age, and abundance versus accretion rates. I appreciate that some "best fit" lines have been produced from this but there is a huge amount of noise and spread in the data for most of the fossil types examined - I think that the authors need to show the r2

values for each line - I suspect that these will be rather low in many cases, and it would be far easier to see the individual trends within sites if each category for each site could be shown. This might be too much for the main text figure but these plots could and should be shown as part of an online supplementary.

Results 3 - The authors talk about and show (Table 1) and specify in the methods that they collected coral constituent data. As a major indicator group and one that is central to the arguments made in the Introduction (indeed the rationale for the study) why is genera/species level coral abundance data not being presented. This is essential and a major and strange omission. It might also go a long way towards explaining some of the vertical accretion rate trends because coral taxa abundance is typically a key driver of reef accretion rates over time and variability therein. The reef geological literature has numerous examples of the links between coral assemblages and accretion rates. I also think that it is inappropriate to link changes in total coral clast abundance (e.g., see first section of discussion) over time without showing/exploring the details of which coral genera/species are changing over time - this is linked to an earlier point about natural ecological changes as the reefs accrete vertically over time. Furthermore, basing coral abundance data on only two narrow (10 cm) core from each site would seem to me to be statistically questionable for a taxa whose dimensions commonly exceed core barrel diameter. greater replication of cores for this type of work are essential and the norm.

Discussion - In light of some of the points made in the discussion that clearly emphasize that some of the reef sites appear to have ceased accreting naturally, the strong and bold statements made in the abstract and in the final paragraph seem too definitive to me. Yes, some reefs may have been altered much earlier than the last few decades, but the record presented here actually appears to show a more complex picture of past natural reef shut down and some evidence of recent slowing. If the weight of opinion of the reviewers is for acceptance I think this is a point that needs to be made more clearly in the abstract and elsewhere in the paper. In the final paragraph the authors also start talking about the impacts of ecological change on coral growth -something they have resented no data for - so this should be removed.

Reviewer #3 (Remarks to the Author):

The manuscript outlines a study that tracks various components in reef cores in Panama and attempts to piece together a sequence of events that took place on the reefs in Panama over the last 3000 years. Two of the cores show that the reef ceased accreting at two of the core sites over 800 years ago, while cores at one site extend to modern. The authors point out that the landscape near the reefs where the cores were extracted has suffered considerable land-use change because of banana agriculture. The authors track herbivorous fish teeth to tease apart changes in herbivorous populations through time. The authors suggest that "the relative abundance of parrotfish teeth is a reliable proxy of absolute parrotfish tooth abundance, as these measures are closely positively correlated (Figure S1)." But the positive correlations suggested in the text are not linear correlations when the

supplementary document is examined. The data instead follow a non-linear, almost tanh function. Changing the function from a positive linear correlation to a non-linear relationship changes the interpretation of the results. Most importantly, however, is that the manuscript suffers from strong undertones that advocate that changes in herbivorous teeth, and therefore fish populations, caused other changes on the reef through time. The authors repeatedly suggested that abundant herbivorous fishes facilitated reef accretion. There is however absolutely no evidence that high densities of parrotfishes helped reef accretion. Herbivorous fish are reef eroders. Changes in hydrography and land-use change may have just as likely shut down the accretion potential at the two sites, and the fishes may have declined, subsequently.

The manuscript by Cramer et al. reminds me of a paper Walbran et al. wrote a paper in Science in 1989, called Evidence from sediments of long-term Acanthaster predation on corals of the Great Barrier Reef. The authors suggested that they could hindcast Acanthaster planci (Crown-of-Thorns seastar, or COT), the seastar, populations using sediment cores. An entire issue of the journal Coral Reefs was dedicated to refuting those findings. For example, Greenstein et al (1992) " ... our results demonstrate the importance of taphonomic processes in altering the original size frequency distribution of the COTS skeleton and their potential for biasing predictions of past population levels derived from constituent particle analyses of surficial reef sediments. " And Pandolfi (1992): "In order to establish a relationship between the number of fossil COTS elements and the original population size, methods must be developed which will relate the number of fossil skeletal elements to the relative abundance of starfish in both the fossil and death assemblages and then to relate the latter to the relative size of the original population."

In summary, the manuscript attempts to track changes in reef accretion in Panama through time and uses proxies for fish, urchin, and foraminifera densities, and then attributes the changes in fish densities to changes in reef accretion rates. This logic is fundamentally flawed. There is plenty of geological literature on taphonomic processes that refute the direct link between core samples and population densities, and there is no evidence that the herbivorous fishes facilitated reef accretion, when modern evidence shows that herbivorous fishes cause the bioerosion of reefs.

Reviewer #4 (Remarks to the Author):

The main finding of the research is that the authors have evidence for a change in the reef community structure towards a declining state cause by historical land clearance and fishing activity. These conclusions rely on the analysis of fossil assemblage changes in multiple reef cores. This finding would be of a broad interest to both reef ecologist and the wider scientific community interested in the longer term anthropogenic impacts on the environment.

There are however some weaknesses in the statistical treatment of the data. Primarily the issue is a conceptual one, in that a number of correlations are extracted from the data linking reef accretion with fossil proxies for the ecosystem state, but there is not definitive

causal link shown. The favoured explanation of the data is that there are changes in the drivers of reef health (water quality, and herbivorous grazing) which then affect the reef accumulation rate and health. But there could be an alternative explanation in that the driver of change is reef accumulation itself, limited by accommodation space or some other parameter, the slowdown in reef accumulation could then change the nature of the habitat (less framework space), limiting the biodiversity of the reef and hence causing the fossil proxies to show a change. I wouldn't necessarily favour either of these explanations only to make the point that the observed correlations do not imply direct causation.

I would recommend that the authors consider in more detail the statistical validity of their correlations, the uncertainties of the timings of the changes in reef state, and the robustness of their causation statements.

The U-Th dating

The age-depth relationships for the 4 cores used in this study are critical for the interpretations that are drawn from the changes in reef assemblages as they define the timing of changes inferred in the reef communities. The U-Th data created for this study is of a high quality and the data are presented in sufficient detail to enable the recalculation of the ages (uncorrected). As is common for young coral U-Th ages a correction is made for unradiogenic ^{230}Th . The method used to do this correction is not the standard approach taken by most coral geochronologists. The method used here is to use a two component contaminant correction rather than the more common single endmember model. The approach used here results in smaller age corrections and hence older ages compared to a more traditional approach. It should be noted that the approach here is supported by the corrected ages being largely in stratigraphic order. That there is one age reversal in the Cayo Adriana core is not unexpected given the potential for corals to not behave as a perfect closed system for U-Th chronology and for there to be some potential variance in the endmember compositions for the two component correction for radiogenic Th. While the data are highly robust the uncertainty in the age model for the core is greater than the analytical precision suggest. This additional uncertainty arises from the imperfections of the coral U-Th system, the potential for uncertainty in the endmember mixing model for ^{230}Th correction, and also for the extrapolation of the ages between dated tie points. It is this last point that will dominate the true age uncertainty of any point in the core. Even with these minor caveats to the age model the data are comfortably good enough to contain the ages of the inferred changes down each core. The greatest part of these age uncertainties comes not from the U-Th data, or the corrections to this data, but comes from the uncertainty in determining the depth in the core where the assemblage data changes. I would not recommend changing the description of the chronology but the authors could consider a short statement outlining where the uncertainty in the age determination of assemblage changes comes from.

The extrapolation of the age model from PD1 (6 U-Th data) to PD2 (2 U-Th data), is not unreasonable but it would be more appropriate to treat them totally separately as the data for the deep parts of the cores do suggest that the age depth relationship for these cores are not totally identical. This shouldn't change materially the final conclusions given the main source of error in the age of the reef state change is in the ecological proxies and not the age models.

The Loess method.

The smoothing parameter chosen, 0.9, is very high limiting the result to show only general trends (as explained in the methods section). Is it possible to shorten the smoothing parameter to show the proposed changes in reef community, if so how sensitive is the result to this?

What degree of polynomial is chosen. In the case of these data even a zero order polynomial would be appropriate as the goal is to find points of change in the moving average and not to determine the rates of change of the assemblages. Regardless of the degree of the local models used the (in)sensitivity of the result to the degree chosen should be assessed (in supplementary information).

The clustering analysis

This is the key test in determining the timing of the proposed changes in assemblages. At present the results of these analyses are not presented very clearly. If the resulting timings of state switch could be shown in a summary figure or table this would improve the communication of the results. For each of the cores. "What is the timing of the switch compared to the accumulation rate changes?" - could be better displayed in a simple figure.

Additionally there is no assessment of the uncertainty in these critical ages derived from the cluster analysis. Is the result sensitive to the clustering method used? A simple way of showing the uncertainty in the ages would be to showing a figure the timings of the 1st, 2nd, and 3rd appearance of each cluster and how much overlap there is between the disappearance of the previous cluster.

Interpretation of the fossil data.

There is potential for bias in the preservation potential of some of the sediment based fossil data. As the reef accumulation rate changes the sedimentary depositional environment will change (less baffling in between coral framework branches, or other changes in the seabed environment). Therefore it is hard to rigorously distinguish changes in fossil assemblage caused by changes in the living community and that caused by changes in the preservation bias.

It is not totally clear how the term "water quality" is being used and how this is derived. As I understand this is derived from the benthic foram assemblage only. What are the levels of accuracy that can be derived from such a foram based proxy reconstruction?

Response to reviewers' comments:

Reviewer #1 (Remarks to the Author):

I read with attention the manuscript entitled "Prehistorical and historical decline of Caribbean reef ecosystems linked to loss of herbivores". This title is very appealing since it suggests a kind of causality. The study is also very timely since the predominant role of herbivores on the maintenance of healthy and productive coral reefs is still very controversial. The main paradigm is certainly the top-down control of the benthos by herbivores and particularly parrotfishes. Yet the other way around is also discussed, and even demonstrated in some cases, with a bottom-up control of herbivore abundance by the benthos (Russ et al. 2015 Mar Biol).

So I was quite excited to see how the authors used historical and even prehistorical records to provide a clear answer to this debate. I must say that I have no personal interest in this debate, I'm just seeking the truth through evidences.

I was very impressed by the originality, quantity and quality of the data. This sampling effort through time and across several groups already makes the paper a strong contribution. However I believe (i) that the manuscript could be simplified and be more straightforward, and (ii) that the conclusions are based on analyses that cannot correctly test the hypothesis and support a causal relationship.

My main criticism is that the authors did not use the appropriate framework and tools to seek causal relationships in time series. The so called "mirage" correlations are very common and I must say the simple plots or GLMs are not convincing. To overcome the pitfalls in the analysis of such trends I suggest to adopt the Granger causality approach (Sugihara et al. 2012 Science). For instance there is a way to test bidirectional vs. unidirectional coupling between variables which would be particularly relevant in the case of herbivore-coral relationships. The authors can also take advantage of their spatial replicates to really try to detect causal relationships (Clarke et al. 2015 Ecology).

- *Thank you for suggesting the causality approaches developed by George Sugihara's group. With direct input from Hao Ye, a researcher in Dr. Sugihara's group who helped to develop the Convergent Cross Mapping (CCM) causality approach, we have leveraged our spatial replication across multiple sediment cores and employed the multispatial CCM approach outlined in Clarke et al. 2015. Our new analyses demonstrate the unidirectional relationship between parrotfish abundance and reef accretion (with the former driving the latter but not vice versa), and eliminate the "mirage" correlations between urchin abundance and accretion.*

The authors should also include more variables in the model like sea surface temperature, which may act on accretion, human density in the vicinity of the reefs, or predator abundance.

- *Unfortunately, sea surface temperature and human population density records are too coarse over space and time during the prehistorical and historical periods to include as predictors of reef accretion. We also do not expect sea surface temperature to have changed in a unidirectional manner over the past few millennia, but fish tooth abundance has steadily declined over the full*

timeseries and reef accretion has declined over latter part of time series. While we are able to track changes in predatory fish abundance from tooth subfossil assemblages, the vast majority of “predatory” tooth morphotypes appear to be from micropredators such as gobies and blennies. Longer-lived top predators such as reef sharks, barracuda, groupers, and snappers are rare in our subfossil record due to their slower turnover rates and/or frequent utilization of non-reef habitat. Details of trends in fish community composition from tooth subfossils will be the focus of a forthcoming separate manuscript.

The manuscript is also too long in my opinion. The last part of the results, based on ordinations, seems useless.

- *The manuscript now focuses on the causal relationships between herbivore (parrotfish and urchin) abundance and reef accretion. Because of the novelty and important ecological and conservation implication of these results, we have removed analysis of trends in other subfossil groups and accompanying ordination and clustering analyses. The paper has been significantly streamlined to focus on the herbivore/accretion relationships, and length has been substantially reduced.*

In sum I believe that this study has the potential to become a key contribution in the field but a more solid analysis of the data is necessary to conclude any causal link and to convince the reader that the loss of herbivores is the cause and not the consequence of the degradation of coral reefs.

- *Thanks to your suggestion, the manuscript now directly demonstrates causality between herbivore loss and declines in reef accretion.*

Reviewer #2 (Remarks to the Author):

This is an interesting paper that builds on previous work by the author on molluscan assemblage changes over time in the same region of the Caribbean. The main take home message about past (historical) declines in reef accretion rates is an important one, although as you will see below I have some major concerns about the interpretations being made and the validity of the datasets given that they derive from individual core records - one from each of 4 geographically isolated reefs.

My main concerns are listed below.

1. The general headline statement about modern reefs and stable alternate states (last line of abstract and discussion). I am not sure this statement about alternative stable states is really accurate (or at least is one that can be supported with the data presented in THIS paper). Indeed, the findings of the paper, if correct, suggest that accretion rates and parrotfish and urchin abundance are correlated. It does not confirm that modern reefs are locked into an alternate stable state. This needs changing.

- *This statement has been removed.*

2. Introduction - Para 1, lines 4-5. These low relief corals are not only tolerant of higher turbidity/nutrients - indeed they occur on many disturbed reefs with little or no turbidity/elevated

nutrient influence. I think it is also the case that they were quite common below/beneath branching coral stands at sites monitored prior to the *Acropora* collapse - it would be worth checking some of the 1970's/early 1980's data from places like Discovery Bay, and have been identified commonly on natural disturbance hiatus surfaces (see Scoffin's classic paper on hurricane deposited sequences).

- *We added a qualifier in this sentence to indicate that Acropora historically dominated in many Caribbean reef locations, but not necessarily in all locations. Indeed, the coral subfossil record from our sediment cores demonstrates that many reefs in Bocas del Toro were historically dominated by non-acroporids. However, those that were dominated by A. cervicornis lost this species at least 50 years ago as water quality declined (see Cramer et al. 2012 Ecology Letters).*

3. Results 1 - I am not entirely convinced about your interpretation of accretion rate trends. Firstly, it is very concerning that accretion rates have been generated based on single cores - especially given that the central thrust of the story being presented hinges entirely on these rates. Accretion rates can commonly vary significantly within and across individual reefs and this raises some major questions to me about the dataset being presented.

- *Budget constraints and the significant expense of generating high-resolution U/Th chronologies did not allow us to assess accretion rate trends for replicate cores within the Airport Point and Cayo Adriana sites. We do show chronologies for two cores within the Punta Donato site. In addition, we do have spatial replication across the broader Bahia Almirante region, including three reef sites that are located at least 10km from each other. Importantly, the revised manuscript focuses on the causal relationship between accretion and herbivory, rather than the absolute trends in accretion through time.*

I note that only one site, Cayo Adriana, actually shows any really convincing reduction in accretion rates in the upper part of the cores (see figs 1 and 2). The other three sites look to have more steady state or even increasing rates over time?

- *Figure 1 (and first paragraph of Results section) show age reversals within the last 500-1000 years represented by the Airport Point and Cayo Adriana cores as well as a slowdown in accretion rates from ~1500-1980 AD in the Punta Donato cores, indicating a slowdown in accretion within the top portions of all cores. Wording in results added to emphasize that the slowdown in accretion is concentrated in the top sections of the cores.*

I note that the authors go on to suggest in the next paragraph that the old age of dates at the tops of some cores is consistent with a natural shut down of reef accretion potential, presumably as accommodation space declined. This would seem to me to be a highly likely interpretation. However, what this also means is it is very hard to assume that the ecological changes you infer are caused by recent environmental factors. It may well be (and I am not sure how you can rule this out - and it rather drives a wedge through the main arguments of the paper) that the up-core changes you infer in all of the cores where actually caused by a natural shift in reef ecology as the reef shallowed i.e., one would expect the types of corals and benthic taxa (especially foraminifera), to change as a reef grows vertically

because the habitats themselves will also be changing. I am not sure why this transition under natural shallowing conditions has not been considered.

- *The paper now focuses on the causal relationship between herbivore abundance and accretion, which is significant across the entire time series and all cores. In the Discussion section, we explicitly state that the antiquity of accretion slowdowns at Airport Point and Cayo Adriana during presumed periods of low human population and anthropogenic reef impacts and the fact that these reefs still have high coral cover in zones adjacent to our coring sites implicates non-human drivers of accretion declines. In contrast, we state that the timing of accretion declines in Punta Donato during a period of intensifying human impacts and the very poor water quality and dearth of living corals today implicate human-caused declines at this site.*

Results 2 - In terms of the measured abundance of fossils in the cores this obviously represents a significant amount of research time and there are some nice plots showing abundance versus age, and abundance versus accretion rates. I appreciate that some "best fit" lines have been produced from this but there is a huge amount of noise and spread in the data for most of the fossil types examined - I think that the authors need to show the r^2 values for each line - I suspect that these will be rather low in many cases, and it would be far easier to see the individual trends within sites if each category for each site could be shown. This might be too much for the main text figure but these plots could and should be shown as part of an online supplementary.

- *As the main focus of the manuscript is on the causal relationship between accretion and herbivore abundance, we have opted to continue to display the temporal trends as raw data points and smoothed trendlines for each core and across all cores. We feel this is the most honest way to show the noise in the data and to avoid imposing linear trends on data that for the most part do not show unidirectional change through time. To allow the raw data to be more prominently displayed, we have reduced the width of the smoothed trendlines.*

Results 3 - The authors talk about and show (Table 1) and specify in the methods that they collected coral constituent data. As a major indicator group and one that is central to the arguments made in the Introduction (indeed the rationale for the study) why is genera/species level coral abundance data not being presented. This is essential and a major and strange omission. It might also go a long way towards explaining some of the vertical accretion rate trends because coral taxa abundance is typically a key driver of reef accretion rates over time and variability therein. The reef geological literature has numerous examples of the links between coral assemblages and accretion rates. I also think that it is inappropriate to link changes in total coral clast abundance (e.g., see first section of discussion) over time without showing/exploring the details of which coral genera/species are changing over time - this is linked to an earlier point about natural ecological changes as the reefs accrete vertically over time.

- *We were initially considering describing trends in coral community composition in a separate manuscript, but have taken your advice and included them here (please see Figure S2). The dominant coral species fluctuates through time within cores and differs across cores, although branching *Porites* is most abundant overall across space and time. Branching *Porites* was the*

only consistently abundant species that could be included in a causal analysis of coral species composition and accretion rate, and this analysis shows no causal relationship between the two. We elaborate on these results in the Discussion section.

Furthermore, basing coral abundance data on only two narrow (10 cm) core from each site would seem to me to be statistically questionable for a taxa whose dimensions commonly exceed core barrel diameter. Greater replication of cores for this type of work are essential and the norm.

- *We agree that a greater number of replicates is always more desirable, but feel that our spatial replication, large number of individual samples (~150), and emphasis on causal relationships rather than absolute temporal trends make our conclusions robust. Additionally, we used Multispatial CCM, which leverages spatial replicates to increase the statistical robustness of results. We would also like to draw your attention to Toth et al. 2015 Nature Climate Change, which drew conclusions about the relationship between historical reef accretion rates and sea surface temperature in the eastern tropical Pacific from a SINGLE core!!*

Discussion - In light of some of the points made in the discussion that clearly emphasize that some of the reef sites appear to have ceased accreting naturally, the strong and bold statements made in the abstract and in the final paragraph seem too definitive to me. Yes, some reefs may have been altered much earlier than the last few decades, but the record presented here actually appears to show a more complex picture of past natural reef shut down and some evidence of recent slowing. If the weight of opinion of the reviewers is for acceptance I think this is a point that needs to be made more clearly in the abstract and elsewhere in the paper. In the final paragraph the authors also start talking about the impacts of ecological change on coral growth -something they have resented no data for - so this should be removed.

- *Thank you for your detailed review – it is much appreciated. We have removed many of the statements about overall environmental change (and their possible drivers), instead focusing on the causal relationship between herbivore abundance and reef accretion.*

Reviewer #3 (Remarks to the Author):

The manuscript outlines a study that tracks various components in reef cores in Panama and attempts to piece together a sequence of events that took place on the reefs in Panama over the last 3000 years. Two of the cores show that the reef ceased accreting at two of the core sites over 800 years ago, while cores at one site extend to modern. The authors point out that the landscape near the reefs where the cores were extracted has suffered considerable land-use change because of banana agriculture. The authors track herbivorous fish teeth to tease apart changes in herbivorous populations through time. The authors suggest that "the relative abundance of parrotfish teeth is a reliable proxy of absolute parrotfish tooth abundance, as these measures are closely positively correlated (Figure S1)." But the positive correlations suggested in the text are not linear correlations when the supplementary document is examined. The data instead follow a non-linear, almost tanh function.

Changing the function from a positive linear correlation to a non-linear relationship changes the interpretation of the results. Most importantly, however, is that the manuscript suffers from strong undertones that advocate that changes in herbivorous teeth, and therefore fish populations, caused other changes on the reef through time. The authors repeatedly suggested that abundant herbivorous fishes facilitated reef accretion. There is however absolutely no evidence that high densities of parrotfishes helped reef accretion. Herbivorous fish are reef eroders. Changes in hydrography and land-use change may have just as likely shut down the accretion potential at the two sites, and the fishes may have declined, subsequently.

- *We now employ causal analyses that definitively show that parrotfish abundance is a positive driver of reef accretion. Interestingly, the causal analyses also demonstrate that the relationship is unidirectional – declines in reef accretion did not cause declines in parrotfish abundance.*

The manuscript by Cramer et al. reminds me of a paper Walbran et al. wrote a paper in Science in 1989, called Evidence from sediments of long-term Acanthaster predation on corals of the Great Barrier Reef. The authors suggested that they could hindcast Acanthaster planci (Crown-of-Thorns seastar, or COT), the seastar, populations using sediment cores. An entire issue of the journal Coral Reefs was dedicated to refuting those findings. For example, Greenstein et al (1992) " ... our results demonstrate the importance of taphonomic processes in altering the original size frequency distribution of the COTS skeleton and their potential for biasing predictions of past population levels derived from constituent particle analyses of surficial reef sediments. " And Pandolfi (1992): "In order to establish a relationship between the number of fossil COTS elements and the original population size, methods must be developed which will relate the number of fossil skeletal elements to the relative abundance of starfish in both the fossil and death assemblages and then to relate the latter to the relative size of the original population."

In summary, the manuscript attempts to track changes in reef accretion in Panama through time and uses proxies for fish, urchin, and foraminifera densities, and then attributes the changes in fish densities to changes in reef accretion rates. This logic is fundamentally flawed. There is plenty of geological literature on taphonomic processes that refute the direct link between core samples and population densities, and there is no evidence that the herbivorous fishes facilitated reef accretion, when modern evidence shows that herbivorous fishes cause the bioerosion of reefs.

- *In the Discussion section, we discuss patterns in our data that suggest minimal taphonomic bias of subfossil abundances. In addition, the conclusions of our revised manuscript are not focused on analysis of absolute population abundance/density but rather on the causal relationship between herbivore abundance and accretion. Lastly, we would like to point out that there is growing evidence for the positive effect of parrotfish herbivory on coral persistence and dominance (including refs cited in manuscript: Mumby Coral reefs 2009, Jackson et al. 2014 GCRMN Report).*

Reviewer #4 (Remarks to the Author):

The main finding of the research is that the authors have evidence for a change in the reef community structure towards a declining state caused by historical land clearance and fishing activity. These conclusions rely on the analysis of fossil assemblage changes in multiple reef cores. This finding would be of a broad interest to both reef ecologists and the wider scientific community interested in the longer term anthropogenic impacts on the environment.

There are however some weaknesses in the statistical treatment of the data. Primarily the issue is a conceptual one, in that a number of correlations are extracted from the data linking reef accretion with fossil proxies for the ecosystem state, but there is not definitive causal link shown. The favoured explanation of the data is that there are changes in the drivers of reef health (water quality, and herbivorous grazing) which then affect the reef accumulation rate and health. But there could be an alternative explanation in that the driver of change is reef accumulation itself, limited by accommodation space or some other parameter, the slowdown in reef accumulation could then change the nature of the habitat (less framework space), limiting the biodiversity of the reef and hence causing the fossil proxies to show a change. I wouldn't necessarily favour either of these explanations only to make the point that the observed correlations do not imply direct causation. I would recommend that the authors consider in more detail the statistical validity of their correlations, the uncertainties of the timings of the changes in reef state, and the robustness of their causation statements.

- *These points are well taken, and we have completed causality analyses that resolve the directional relationships between herbivore abundance/coral community composition and accretion rates. These analyses also avoid “mirage” or spurious correlations (please see reviewer #1’s comments and our responses).*

The U-Th dating

The age-depth relationships for the 4 cores used in this study are critical for the interpretations that are drawn from the changes in reef assemblages as they define the timing of changes inferred in the reef communities. The U-Th data created for this study is of a high quality and the data are presented in sufficient detail to enable the recalculation of the ages (uncorrected). As is common for young coral U-Th ages a correction is made for unradiogenic ^{230}Th . The method used to do this correction is not the standard approach taken by most coral geochronologists. The method used here is to use a two component contaminant correction rather than the more common single endmember model. The approach used here results in smaller age corrections and hence older ages compared to a more traditional approach. It should be noted that the approach here is supported by the corrected ages being largely in stratigraphic order. That there is one age reversal in the Cayo Adriana core is not unexpected given the potential for corals to not behave as a perfect closed system for U-Th chronology and for there to be some potential variance in the endmember compositions for the two component correction for radiogenic Th. While the data are highly robust the uncertainty in the age model for the core is greater than the analytical precision suggest. This additional uncertainty arises from the imperfections of the coral U-Th system, the potential for uncertainty in the endmember mixing model

for ^{230}Th correction, and also for the extrapolation of the ages between dated tie points. It is this last point that will dominate the true age uncertainty of any point in the core. Even with these minor caveats to the age model the data are comfortably good enough to contain the ages of the inferred changes down each core. The greatest part of these age uncertainties comes not from the U-Th data, or the corrections to this data, but comes from the uncertainty in determining the depth in the core where the assemblage data changes. I would not recommend changing the description of the chronology but the authors could consider a short statement outlining where the uncertainty in the age determination of assemblage changes comes from.

The extrapolation of the age model from PD1 (6 U-Th data) to PD2 (2 U-Th data), is not unreasonable but it would be more appropriate to treat them totally separately as the data for the deep parts of the cores do suggest that the age depth relationship for these cores are not totally identical. This shouldn't change materially the final conclusions given the main source of error in the age of the reef state change is in the ecological proxies and not the age models.

- *Thank you for thinking about and enumerating potential age estimation uncertainties in such detail. We have added a brief section in the Discussion that explicitly addresses potential sources of uncertainty.*

The Loess method.

The smoothing parameter chosen, 0.9, is very high limiting the result to show only general trends (as explained in the methods section). Is it possible to shorten the smoothing parameter to show the proposed changes in reef community, if so how sensitive is the result to this?

- *Is it possible to do this, but we feel it would not be very informative because the focus of the revised paper is on the causal relationship between herbivore abundance and accretion rates. The temporal trends shown in Figure 3 are merely descriptive – no statistics have been done. We have opted to show the raw data with points, and to guide the readers' eye to overall trends with the highly smoothed lines.*

What degree of polynomial is chosen. In the case of these data even a zero order polynomial would be appropriate as the goal is to find points of change in the moving average and not to determine the rates of change of the assemblages. Regardless of the degree of the local models used the (in)sensitivity of the result to the degree chosen should be assessed (in supplementary information).

- *For reasons outlined immediately above and in responses to "Results 2" comment of Reviewer 2 and to keep the focus of the paper streamlined, we opted to not draw straight lines through the timeseries data.*

The clustering analysis

This is the key test in determining the timing of the proposed changes in assemblages. At present the results of these analyses are not presented very clearly. If the resulting timings of state switch could be shown in a summary figure or table this would improve the communication of the results. For each of

the cores, "What is the timing of the switch compared to the accumulation rate changes?" - could be better displayed in a simple figure. Additionally there is no assessment of the uncertainty in these critical ages derived from the cluster analysis. Is the result sensitive to the clustering method used? A simple way of showing the uncertainty in the ages would be to showing a figure the timings of the 1st, 2nd, and 3rd appearance of each cluster and how much overlap there is between the disappearance of the previous cluster.

- *Ordination and cluster analyses have now been removed.*

Interpretation of the fossil data

There is potential for bias in the preservation potential of some of the sediment-based fossil data. As the reef accumulation rate changes the sedimentary depositional environment will change (less baffling in between coral framework branches, or other changes in the seabed environment). Therefore it is hard to rigorously distinguish changes in fossil assemblage caused by changes in the living community and that caused by changes in the preservation bias.

- *These are valid points. We have added a statement in the Discussion addressing the potential effects of differential baffling from varying accretion rates.*

It is not totally clear how the term "water quality" is being used and how this is derived. As I understand this is derived from the benthic foram assemblage only. What are the levels of accuracy that can be derived from such a foram based proxy reconstruction.

- *Water quality proxy from foram data has now been removed, and paper focuses on herbivore/accretion causal relationship.*

Reviewers' comments:

Reviewer #1 (Remarks to the Author):

I appreciate the attention paid by the authors to our comments. A huge work has been done to make the results more convincing and to reinforce the causality in the relationships.

Overall this work is now a strong contribution to the field.

I still have some comments:

- The abstract should gain to provide more methodological details about causality analysis. The paper can be read and cited only for that aspect. Line 28 "detailed ecological Baseline" is too vague and some colleagues can get upset.
- Line 53 ref 1-4 should be upper case
- Line 80 "andreef" ?
- At the beginning of the results a short description of sites and samples is necessary without going to the methods.
- Line 118 split the sentence after fragments.
- Figure 4 clearly say what is sample size. First panel why .0.9? Pearson and p-values are too confusing, please change.
- Some recent references need to be cited and discussed on this topic like Bruno1 & Valdivia (2016), Zaneveld (2016), Renema (2016), but above all Bozec (2016).

Reviewer #3 (Remarks to the Author):

The manuscript has been greatly improved. The authors have included the CCM causality approach to rigorously test the likely relationships between reef accretion rates and biological proxies. The earlier version of the manuscript had no such rigor. The authors have addressed all my concerns except one. In their reply they state on Page 6: "the conclusions of our revised manuscript are not focused on analysis of absolute population abundance/density but rather on causal relationship between herbivore and [reef] accretion". But in the Abstract, they write "Causality analyses revealed that accretion rates were driven by parrotfish abundance (but not vice versa)". In fact, the authors state throughout the manuscript, particularly in the Discussion section, that they were linking accretion with parrotfish abundance, for example, on Page 11, lines 247 "reef accretion may cease on many reefs if parrotfish abundance remains low." It seems that the authors cannot scale up their findings of a few paleo-teeth to reflect past densities of parrotfish populations, and they need to add a strong caveat pointing to this considerable leap of faith.

Reviewer #4 (Remarks to the Author):

The authors have substantially improved on the manuscript following the last round of review. The conclusion that the decline in the parrot fish abundance is robustly shown to be responsible for the decline in the reef accretion is a novel and important finding that merits publication. I am still a little unconvinced on the arguments surrounding the relative timings of the changes in the ecological assemblages. These relative timings however are not crucial to the arguments of causality between the ecological changes and the reef accretion rates. My main concern is that the uncertainties of the timings of the changes are not dominated by the U-Th chronology but are more substantially driven by the high degree of scatter in the assemblage data. I would recommend that the authors be encouraged to reconsider the statements surrounding lines 192-204 as these are not central arguments, and the accuracy of the timings of the changes is not demonstrated to be good enough. The main conclusions of the causality of the reef decline is however strong and should be published.

Response to reviewers' comments:

Reviewer #1 (Remarks to the Author):

I appreciate the attention paid by the authors to our comments. A huge work has been done to make the results more convincing and to reinforce the causality in the relationships.

Overall this work is now a strong contribution to the field.

I still have some comments:

- The abstract should gain to provide more methodological details about causality analysis. The paper can be read and cited only for that aspect. Line 28 "detailed ecological Baseline" is too vague and some colleagues can get upset.

- *First off, thank you for your extremely helpful and insightful suggestions. They have enabled us to take our results further than we initially envisioned.*
- *We added wording to abstract indicating we tested cause and effect relationships using convergent cross mapping. We did not want to add to many methodological details to abstract, as it is now just over the 150 word limit and convergent cross mapping methods have been detailed quite well in the recent literature by the researchers that have developed these techniques. We also replaced "detailed ecological baseline" with "quantitative ecological data prior to large-scale human impacts".*

- Line 53 ref 1-4 should be upper case

- *Not quite sure what you are referring to here.....*

- Line 80 "andreef" ?

- *Corrected to "and reef"*

- At the beginning of the results a short description of sites and samples is necessary without going to the methods.

- *Moved brief description of sites from beginning of Methods to beginning of Results section. We decided to leave information about sampling of cores in the Methods section per Nature Communications formatting guidelines.*

- Line 118 split the sentence after fragments.

- *Done.*

- Figure 4 clearly say what is sample size. First panel why .0.9? Pearson and p-values are too confusing, please change.

- *In the Methods section of text and Figure 4 axis and legend we changed “sample size” to “number of core samples”. We corrected .0.9 to 0.9. We also removed rho (ρ) from Pearson correlation coefficient to avoid confusion with p , and now just refer to test statistic as Pearson correlation coefficient.*

- Some recent references need to be cited and discussed on this topic like Bruno1 & Valdivia (2016), Zaneveld (2016), Renema (2016), but above all Bozec (2016).

- *Thank you for these great suggestions. We have incorporated all papers into the revision but that of Renema 2016. Although this is an excellent paper, it is not immediately relevant as it deals with global Acropora response to rapid sea level fluctuations that likely largely ceased prior to 3,000 years ago.*

Reviewer #3 (Remarks to the Author):

The manuscript has been greatly improved. The authors have included the CCM causality approach to rigorously test the likely relationships between reef accretion rates and biological proxies. The earlier version of the manuscript had no such rigor. The authors have addressed all my concerns except one. In their reply they state on Page 6: "the conclusions of our revised manuscript are not focused on analysis of absolute population abundance/density but rather on causal relationship between herbivore and [reef] accretion". But in the Abstract, they write "Causality analyses revealed that accretion rates were driven by parrotfish abundance (but not vice versa)". In fact, the authors state throughout the manuscript, particularly in the Discussion section, that they were linking accretion with parrotfish abundance, for example, on Page 11, lines 247 "reef accretion may cease on many reefs if parrotfish abundance remains low." It seems that the authors cannot scale up their findings of a few paleo-teeth to reflect past densities of parrotfish populations, and they need to add a strong caveat pointing to this considerable leap of faith.

- *Thanks for your comments, which have vastly improved our paper. We feel confident that tooth assemblages were not affected by preservational biases that vary through time (thoroughly addressed in Discussion section), and that the fish tooth subfossil record is just as reliable as the coral and urchin subfossil records. Fortunately, fish teeth are quite abundant in our reef cores (we added a sentence in Results that states the mean number of teeth per sample = 74 and range = 2-232), allowing for rigorous statistical analyses of trends in tooth abundance. Parrotfish made up about half of the tooth assemblage, so analyses of parrotfish tooth trends are based on adequate sample sizes. Further, we conducted causality analyses of both absolute AND relative abundance of parrotfish teeth and found that BOTH measures positively affected accretion rates. We do acknowledge that the tooth subfossil record likely overestimates the contribution of parrotfish to the total living reef fish community, and have added a statement detailing this in the Discussion.*

Reviewer #4 (Remarks to the Author):

The authors have substantially improved on the manuscript following the last round of review. The conclusions that the decline on the parrot fish abundance is robustly shown to be responsible for the

decline in the reef accretion is a novel and important finding that merits publication. I am still a little unconvinced on the arguments surrounding the relative timings of the changes in the ecological assemblages. These relative timings however are not crucial to the arguments of causality between the ecological changes and the reef accretion rates.

My main concern is that the uncertainties of the timings of the changes are not dominated by the U-Th chronology but are more substantially driven by the high degree of scatter in the assemblage data. I would reconvened that the authors be encouraged to reconsider the statements surrounding lines 192-204 as these are not central arguments, and the accuracy of the timings of the changes is not demonstrated to be good enough. The main conclusions of the causality of the reef decline is however strong and should be published.

- *Thank you for your comments, which have vastly improved our paper. We feel that the extreme analytical precision of the U-Th dates ($\pm 3 - 15$ years), the large number of dates obtained, and the lack of age reversals in our chronology (except where accretion slows down in core tops, a caveat which we state in the Discussion) justifies us making inferences about the broad timing of changes. We avoided placing emphasis on the exact timing of changes, and only relate our chronology to general trends in human population and human activities within the span of a century or so. Lastly, accretion rates are based on this chronology, and we find a clear causal relationship between parrotfish abundance and accretion which has been hypothesized by numerous other researchers. We did add the "approximate" sign (~) before all of our dates in this second revision to emphasize that these years are not exact.*

REVIEWERS' COMMENTS:

Reviewer #3 (Remarks to the Author):

One final suggestion is to change the time axis of Figure 3. It is not conventional to describe time in AD. It is best to describe time as years before present (BP); for example 2000 cal. BP.

RESPONSE TO REVIEWERS' COMMENT:

Reviewer #3 (Remarks to the Author):

One final suggestion is to change the time axis of Figure 3. It is not conventional to describe time in AD. It is best to describe time as years before present (BP); for example 2000 cal. BP.

- To reflect the focus of our study – reef ecological change from the prehistorical period to the present – we intentionally used year AD (the convention used in ecological and historical studies) rather than year BP (typically used in geological studies). This study is mainly geared towards ecologists and conservation practitioners, so we feel that year AD is more appropriate.